# Emulsion Polymerization Using an Amphiphilic Oligoether Ionic Liquid as a Surfactant

**DOI:** 10.3390/polym14173475

**Published:** 2022-08-25

**Authors:** Ariadna Jiménez-Victoria, René D. Peralta-Rodríguez, Enrique Saldívar-Guerra, Gladis Y. Cortez-Mazatán, Lluvia de Abril A. Soriano-Melgar, Carlos Guerrero-Sánchez

**Affiliations:** 1Centro de Investigación en Química Aplicada (CIQA), Blvd. Enrique Reyna #140, 25294 Saltillo, Mexico; 2Laboratory of Organic and Macromolecular Chemistry (IOMC), Jena Center for Soft Matter (JCSM), Friedrich Schiller University Jena, 07743 Jena, Germany

**Keywords:** ionic liquid C1EG™, emulsion polymerization, ionic liquid surfactant, styrene, methyl methacrylate, colloidal polymeric dispersion

## Abstract

We investigate the use of an ionic liquid (IL) as a surfactant in emulsion polymerization (EP) reactions. ILs have been proposed as surfactants for micellar dispersions, emulsions, micro-emulsions and suspensions. Thus, it is important to acquire knowledge of the application of ILs in heterogeneous polymerizations. We selected the amphiphile cationic oligoether IoLiLyte C1EG™ as an IL for this purpose and compared its performance to that of the conventional surfactant dodecyl trimethyl ammonium bromide (DTAB) in the EP of methyl methacrylate and styrene. After we found the proper concentration range of the IL, this amphiphile showed similar polymerization rates to those observed with DTAB for both monomers. The evolution of monomer conversion and the final average diameter of formed polymeric particles were similar for both evaluated surfactants, demonstrating their capability to stabilize the EPs of the investigated monomers. We simulated the evolution of monomer conversion and particle size using a conventional model for emulsion polymerization, which showed good agreement with the experimental data, suggesting that the EP with this IL follows Smith-Ewart kinetics.

## 1. Introduction

Ionic liquids (ILs) are organic salts with a melting point below 100 °C, and they continue to attract much interest in many fields of chemistry and industry due to their chemical and physical properties [1,2]. ILs are extremely versatile; their properties can be easily adjusted by varying the cationic and anionic species in their chemical structure [3]. As solvents, they generally show negligible vapor pressure, high thermal stability, and the ability to solubilize a wide range of compounds [4,5]. These properties make them interesting alternatives to traditional organic solvents, and they have already found use in organic/inorganic synthesis [6], catalysis [7,8], electrosynthesis [9,10], and polymer synthesis [11,12,13]. The use of ILs as media to perform diverse polymerizations, including conventional free radical, living/controlled radical, ionic, and coordination polymerizations has been proposed and reviewed by several research groups [14,15,16,17,18]. One of the properties of ILs that has attracted interest is their ability to form nanostructures [4,19,20]. Hence, ILs have been proposed as surfactants for micellar dispersions, emulsions, and microemulsions; however, knowledge in these applications is still significantly limited [21,22,23]. For instance, imidazolium-based ILs can show amphiphilic nature due to their *n*-alkylimidazolium cations, which may promote interfacial and/or aggregation (self-assembly) behavior in aqueous media and lead to micelle formation [18,19]. Changes in the length of the alkyl chain and/or the type and size of anion influence the dynamics of formed micellar aggregates and corresponding critical micelle concentration (CMC) [24,25]. Thus, it is thought that this type of IL might show behavior similar to that of conventional surfactants in aqueous media, which makes them interesting alternatives to stabilize heterogeneous polymerization reactions. In suspension polymerization, ILs can act as stabilizing agents by inducing electrostatic charges on the surfaces of monomer droplets and polymer particles to prevent coalescence, and depending on the type and concentration of ILs, the particle size of the formed polymer beads can be modulated [26]. In addition, ILs have also been used as initiators of polymerization reactions in aqueous media, for instance, in the cationic emulsion polymerization (EP) of styrene (St) [27].

Concerning the use of ILs as surfactants, Costa et al. [28] utilized 1-*n*-dodecyl-3-methylimidazolium chloride, [C12mim]Cl (CMC = 16.7 mM), to stabilize the EP of methyl methacrylate (MMA) and compared the results to those obtained from the use of dodecyltrimethylammonium bromide (DTAB; CMC = 12.6 mM) in similar experiments. This demonstrated the feasibility of using ILs as stabilizers in EPs, as monomer conversion, average particle size, and average molecular weights were similar to those obtained with a conventional surfactant.

Other reports describe the use of ILs or their derivatives (including polymerized ILs) as surfactants in micro/mini-EP processes, but mainly from a formulation perspective [29]. Similarly, the use of ILs as surfactants for micro/mini EPs has been reported for free-radical polymerization [30,31,32,33], AGET-ATRP (activators generated by electron transfer-atom transfer radical polymerization) of MMA [34,35,36], and for reversible addition-fragmentation chain transfer (RAFT) polymerization of St [37], allowing access to latexes of monodisperse nanoparticles and good colloidal stability. In another application, Rozik et al. prepared nanoparticles from a polymerizable IL (vinylimidazolium-based) and used them to stabilize the EP of St; interestingly, the initial hydrophilic character of the stabilizer turns hydrophobic when the latex is dried, forming films [38]. Similarly, a conjugated polyfluorene has been used in combination with dialkylimidazolium bromide (IL) to prepare nanoparticles capable of stabilizing the EP of St for obtaining stable and luminescent latexes [39]. Finally, Sui et al. also reported the use of poly(ferrocenylsilane)-based poly(IL)s (PFS-PILs) as stabilizers in the micro-EP of MMA [40].

IoLiLyte C1EG™ (formerly known as TEGO™ IL K5MS or AMMOENG™100) [20,41] belongs to a family of oligoether-based ILs; it is a readily available amphiphilic acyclic ammonium salt-containing cation based on oligo(ethylene glycol) units with different chain lengths and a hydrophobic segment (i.e., long alkyl side chain), and it is currently commercialized by IoLiTec GmbH at reasonable prices, turning this IL into an interesting alternative to other, more-expensive imidazolium-based ILs [10,23,24,39,41]. Next to other oligoether-based ILs and based on the abovementioned properties, C1EG™ (C1EG in the rest of this paper) has drawn interest for industrial applications such as high-performance additives and dispersants in the areas of coatings, lacquers, and inks [41].

EP is a well-established industrial process for polymer dispersions and is crucial for modern life in many fields [42,43,44]. The colloidal nature of EP and polymer dispersions is the source of all benefits and drawbacks related to their applications [44,45]. The large number of small particles present during an aqueous EP is technically beneficial because it allows rapid polymerization and efficient heat removal. Additionally, EP is considered clean and environmentally friendly because it is carried out in aqueous dispersion media, as opposed to other processes that generate residual volatile organic compounds (e.g., solution polymerization in organic solvents). However, the main disadvantage of EP relates to the purity of the produced polymers, which contain residues (up to 5 wt. %) of auxiliary materials, mainly surfactants. All surfactants for aqueous EP are, according to Bancroft’s rule [46], hydrophilic, and their presence in the final polymer increases the hydrophilicity of the system throughout the intended application. In addition, low molecular weight surfactants can easily desorb or migrate under changing conditions. In a polymer latex, this often leads to the formation of undesirable foam and the harmful absorption of water during the application of the polymer. These facts have driven, in the last decades, the search for strategies to overcome the current limitations of conventional surfactants [44,47,48]. For instance, it is common practice to use modern water-based universal pigment pastes (also called colorants) to tint every kind of water-based coating or lacquer, e.g., flat and high-gloss acrylics, poly(vinyl acetate) (PVAc) based paints, or alkyd emulsion-based systems, where the use of ILs has enabled universal colorants to be applied extensively [41].

Hence, the main purpose of this contribution is to investigate the EP of St and MMA using IL C1EG as a surfactant and compare its performance to that of DTAB (a conventional ionic surfactant) and another IL (1-*n*-dodecyl-3-methylimidazolium chloride, ([C12mim]Cl)), using, in the latter case, the experimental data reported by Costa et al. [28] The polymerization reactions were also simulated using a conventional mathematical model for EP implemented in the POLYRED package [49], which acceptably predicted the obtained experimental data, confirming that the standard assumptions of conventional EP models are also applicable to CIEG-stabilized EPs. In addition, some physical properties of films derived from the prepared polymer latexes were determined and compared to gain better understanding of the influence of C1EG as an additive on potential applications such as coatings and inks.

## 2. Materials and Methods

The monomers used were styrene (St) and methyl methacrylate (MMA) (Sigma-Aldrich 99% purity for each monomer, Saint Louis, MO, USA). St and MMA were passed through a column (Sigma-Aldrich, Saint Louis, MO, USA) to remove the inhibitors. V-50 (2,2’-azobis(2-methylpropionamidine) dihydrochloride) from Sigma-Aldrich (97% purity, Saint Louis, MO, USA) was used as initiator. The surfactants to stabilize the EPs were C1EG (>95% purity) from IoLiTec (Heilbronn, Baden-Wurtemberg, Germany) and dodecyltrimethylammonium bromide (DTAB, ≥98% purity) from Sigma-Aldrich (Saint Louis, MO, USA). Both surfactants were used without further purification. Nitrogen gas (N_2_, 99.99% purity, Infra, Saltillo, Mexico) was used to deplete oxygen from the reaction systems and to maintain an inert atmosphere in the reactor. In all reactions, purified and deionized water was used (Milli-Q filtration system, Molsheim, Bas-Rhin, Grand Est, France). The inhibitor to stop the EP reactions in the withdrawn aliquots was hydroquinone (Aldrich, 98% purity; Saint Louis, MO, USA), 1% *w*/*w* aqueous solution. Figure 1 shows the molecular structure of the C1EG, DTAB, and [C12mim]Cl cationic surfactants.

### 2.1. Experimental Section

The critical micellar concentration (CMC) of IL IOLILYTE C1EG in water was determined by the bubble rising method in a Sensa-Dyne tensiometer (Model PCL 500, Sensa-Dyne, Mesa, AZ, USA) at 25 °C.

The formulations used for the EP reactions and some results derived thereof are summarized in Table 1. The nomenclature used for the reactions shows a non-consecutive numeration; we decided to do so to keep the references consistent with the original experiments in the laboratory notebooks. EPs (in duplicate or triplicate in some cases) were carried out in batch mode: initiator, water, and surfactant were added to a 50 mL glass reactor provided with a three-mouth lid, as depicted in Figure 2, and equipped with a condenser (Sigma-Aldrich, Saint Louis, MO, USA), mechanical stirrer (Rawang, Selangor, Malasia), thermocouple (K Type, Omega Engineering, Norwalk, CT, USA), and a stainless-steel needle (Popper & Sons, New Hyde Park, New York, USA) for N_2_ sparging. N_2_ was sparged into the reaction mixture under stirring at 300 rpm for 2 h prior to the reaction. When the reactor mixture reached the temperature set-point, 70 °C, the oxygen-free monomer (obtained by sparging N_2_ into a separate flask containing the respective monomer for a few minutes) was loaded into the reactor with a single shot and time zero was noted. The reaction temperature was controlled (±1 °C) by using a constant-temperature water/thermal oil circulating bath (Polyscience 9010, Polyscience, Niles, IL, USA) connected to the reactor jacket. Aliquots were withdrawn from the reaction mixture after 10, 20, 30, 60, 90, and 120 min for gravimetric determination of monomer conversion and particle size analysis. The aliquots were placed in vials containing an aqueous solution of hydroquinone (0.5 g) to stop the polymerization. All reactions were carried out for 120 min (total time).

### 2.2. Characterization

Monomer conversions were determined by gravimetric analysis. Latex samples were withdrawn from the reactor and placed into pre-weighed vials containing a small (known) amount of hydroquinone solution of known concentration. Thereafter, the vials were weighed again and stored at −15 °C. After 24 h, the samples were lyophilized and weighed again to estimate monomer conversion. The average intensity diameter and particle size distribution of the polymer particles were determined by dynamic light scattering (DLS) using a Malvern Zetasizer^®^ model Nano S90 (Malvern, Worcestershire, United Kingdom). For this analysis, the latex samples were diluted 1:10 with Milli-Q water (Molsheim, Bas-Rhin, Grand Est, France).

The number of polymer particles was calculated from the results of monomer conversion and the average diameter of the particles.

The molecular weight distributions of the produced polymers were determined by gel permeation chromatography (GPC) in an Agilent GPC system (model G7810A) equipped with a PLGEL MIXED-C (5 μm) column (Agilent, Santa Clara, CA, USA) using tetrahydrofuran (THF, Sigma-Aldrich, Saint Louis, MO, USA) as an eluent (0.1% *w*/*v*). Polymer samples for GPC analysis were prepared in THF for each lyophilized material.

### 2.3. Contact Angle (CA) of Polystyrene (PSt) Films

Three approaches were followed to determine the contact angle of the polystyrene films. In the first approach, the respective PSt latex dispersion (10 mL), as obtained at the end of the reaction, was freeze-dried until it reached a constant weight. In the second approach, another 10 mL PSt dispersion sample was freeze-dried and subsequently washed with hot water to remove residual surfactant until no foam formation was visually detected. This sample was then freeze-dried until it reached a constant weight. In the third approach, 10 mL of the respective PSt latex dispersions was thoroughly washed with hot water until the value of the electric conductivity of the residual water was very close to that of pristine water (Milli-Q). For CA determination, 1.8 g samples of freeze-dried PSt were slowly dissolved (to avoid bubble formation in the samples with surfactant) in 7.5 mL of THF. This solution was cast onto a Petri dish and protected from dust contamination, and THF was evaporated overnight to obtain the corresponding polymer film. Finally, the CA was determined in an NRL CA goniometer (Ramé-Hart, Model 100-00; Mountain Lakes, NJ, USA), three repetitions. The purpose of these tests was to determine the CAs of PSt films with the surfactant at the end of the reaction, residual surfactant after partial washing, and without surfactant.

### 2.4. Mathematical Modeling

The EPs were simulated using the mathematical model implemented in the simulation package POLYRED (University of Wisconsin, Madison WI, USA). The core of the POLYRED model is a population balance equation that describes the quantity *F(m,t)* represented in Equation (1), where *F(m,t)dm* denotes the number of particles per liter of water in the latex having a polymer mass (a measure of particle size) between *m* and *m + dm* at reaction time *t*.
(1)$$\frac{\partial F\left(m,t\right)V_{w}}{\partial t}+\frac{\partial V_{w}F\left(m,t\right)\frac{dm}{dt}}{\partial m}=G-D$$
where *V_w_* is the volume of water in the latex, and *G* and *D* represent, respectively, all the terms contributing to the generation or disappearance of particles having a polymer mass *m*, such as nucleation and coalescence terms (zero in the batch case).

Equation (1) has the following boundary condition (Equation (2)), which represents the micellar nucleation of particles due to the entry of a radical from the aqueous phase to the micelles (regarded as particles of micellar size *m_m_*):(2)$$V_{W}F\left(m,t\right){\left.\frac{dm}{dt}\right|}_{m_{m}}=V_{aq}N_{A}\left(a_{m}k_{m_{m}}M{\left[P\right]}_{w}+a_{m}k_{m_{mR}}M{\left[R\right]}_{w}\right)$$
where *V_aq_* is the aqueous phase volume, a_m_ is the surface area of a micelle, *M* is the micellar concentration resulting from the added surfactant, $k_{m_{m}}$ and $k_{m_{mR}}$ are, respectively, entry rate coefficients for growing oligomeric radicals with concentration ${\left[P\right]}_{w}$ or primary radicals with concentration ${\left[R\right]}_{w}$ in the aqueous phase, and *N_A_* is Avogadro’s number.

This model assumes an average number of radicals $\bar{n}$ in all the particles of a given size provided by the Stockmayer–O’Toole solution in terms of Bessel functions of the Smith–Ewart recurrence equation [50,51], which includes entry, desorption, and termination of radicals in the particle. The model is complemented by mass balances for the monomer, monomer bound in polymer, initiator, surfactant, and all the reacting species, in the form of ordinary differential equations (ODEs), as well as the thermodynamic partitioning of monomers between aqueous, droplet, and particle phases present in the emulsion via partition coefficients (algebraic equations). The partial differential Equation (1) is discretized along the variable *m* using orthogonal collocation on finite elements yielding ODEs, so the whole model results in a set of differential–algebraic equations that is numerically solved in POLYRED once the kinetic, physico–chemical, and experimental parameters of the system are fed to the software. See Saldívar and Ray [52] for more details.

### 2.5. Methodology Used in the Quantitative Analysis of Results

Different tools of analysis were used to interpret the experimental results. NCSS software (NCSS 2007; LLC Kaysville, UT, USA) for ANOVA and post hoc Tukey’s test, for multiple comparisons of means (pair by pair) were used for each monomer to detect statistically significant differences (*p* ≤ 0.05) between treatments and to assess the effects of variables (initiator and surfactant concentrations and type of surfactant) on the studied responses (essentially monomer conversion and average particle size). This tool can also be used to study gross global effects of the main variables, and in this work, it was applied to study the overall effect of the surfactant type on the main responses, ignoring the possible effects of other variables. Although some statistical confounding may arise from using this type of analysis, major effects are made evident in this way. In another type of analysis, smaller subsets of experiments were compared to avoid confounding between independent variables, taking advantage of the factorial nature of the original experimental design. Pairs of experiments in which one of the variables was changed while the other variable was maintained constant were compared; these comparisons were rounded out by the ANOVA outcome. The analysis of results was finally complemented using a mechanistic point of view, i.e., by simulations of the EPs using the simulation package POLYRED for polymerization reactors, and by comparing the simulation results with the obtained experimental data.

## 3. Results and Discussion

The CMC of IL C1EG was determined as 2.247 mM (0.1354 wt. %), which is in good agreement with the value reported by the supplier: 2.132 mM (0.1285 wt. %) [20]. The observed difference may be ascribed to the different experimental techniques.

It has been reported that the CMC of ionic surfactants in aqueous media decreases with increasing ionic strength of the solution [53], as the presence of salts might promote electrostatic repulsion between charged moieties, facilitating the aggregation of surfactant molecules. Taking into account the lower hydrophilicity of IL ions (as compared to the common inorganic ions), it is expected that the CMC of ionic surfactants would have a lower value due to a stronger union of the counter-ion as the water produces less resistance to IL ions.

Initial EPs carried out with C1EG as surfactant using concentrations of 2–3-fold its CMC were not completely satisfactory because the monomer conversions reached values of only 74% in 8 h. Due to this, the concentration of C1EG was increased until monomer conversion reached over 90% after 2 h. The concentration of C1EG finally used was in the range of 2.1–2.9 × 10^−2^ mol L^−1^ (12.68–17.51 g L^−1^; 7.7–11-fold its CMC). From a practical point of view, the selection of C1EG as a surfactant in a commercial application would mainly depend on economic and environmental factors, as well as on the colloidal stability of the latex. On a purely economic basis, both the necessary amount and unit price of the IL as surfactant would determine its viability for commercial use. However, the amount of surfactant necessary would have to be pre-established based on desired polymerization rate (productivity) and colloidal stability.

Table 1 summarizes the obtained experimental results (i.e., monomer conversion, latex average particle diameter, number average molecular weight, dispersity of the polymers at final monomer conversion, and number of particles) of the performed EPs of the two monomers, St and MMA, and varying concentrations of C1EG, DTAB, and initiator. Experimental results for individual EPs are also shown in Appendix A.

Before proceeding to the analysis and discussion of the experimental results of the reactions, some additional details regarding the sequence of presentation are provided. First, the global effect of the type of surfactant is analyzed for monomer conversion, particle size, and molecular weight, mainly by using ANOVA and the Tukey’s comparison test. Thereafter, smaller subsets of experiments are analyzed in more detail to understand the main effects of changing one individual variable at a time (surfactant or initiator concentration) while other factors are kept constant. These analyses are supported by complementary ANOVA comparisons and POLYRED simulations.

### 3.1. Global Effect of Surfactant Type

Figure 3 and Figure 4 show the evolution of monomer conversion with reaction time for St and MMA EPs, respectively. For St (Figure 3), there is no clear difference in the final monomer conversion for the reactions performed with the two surfactants, which is confirmed by ANOVA (Appendix A). For MMA (Figure 4), consistent with the ANOVA results (Appendix A), reactions carried out in the presence of IL C1EG exhibited slightly higher monomer conversion as compared to that of DTAB; likewise, higher stability in the emulsion is observed (visually assessed), although the higher stability could also be ascribed to the relatively higher concentration of C1EG with respect to DTAB (on a molar basis).

#### 3.1.1. Styrene (St)

Appendix A shows the results of ANOVA and post hoc Tukey’s test for the global effect of the surfactant type on the EPs of St. There were no significant differences in average monomer conversion when using the two surfactants (89.69% ± 5.16% for C1EG and 92.03% ± 3.77% for DTAB). However, differences were found in the average particle diameter (*D*_p_): 66.34 ± 2.51 nm for C1EG and 58.04 ± 3.50 nm for DTAB. These differences in *D*_p_ could be ascribed to the larger size of the C1EG molecule compared to the size of the DTAB molecule, as depicted in Figure 1. Increases in *D*_p_ of micelles (precursors of polymer particles), as affected by the size of the surfactant molecule, have been documented before. UsYarov et al. [54] investigated the influence of a homologous series of cationic surfactants with increasing alkyl chain size (alkyltrimethylammonium bromides containing 10, 12, 14, and 16 carbon atoms in their alkyl chains), including DTAB (12 carbon atoms), on the size of micelles, and found that *D*_p_ increases from 4.24 nm for decyltrimethylammonium bromide up to 7.44 nm for hexadecyltrimethylammonium bromide, whereas for DTAB they found *D*_p_ values of 5.00 nm. Micelles, as precursors of polymer particles, are expected to grow as a result of incorporating monomers followed by radical polymerization in an EP. As per *D*_p_ variation of polymeric latex particles, Fernandez and Jebbanema [55] studied the effect of different anionic surfactants on the properties of an acrylic terpolymer prepared by EP. They used a series of Disponil FES ethoxylated (EO) surfactants (0 to 50 EO units) and found that starting from zero EO (*D*_p_ = 304 nm), there is a minimum of two to four EO units required to generate “large” particles (*D*_p_ = 235 and 240 nm, respectively); *D*_p_ increased to a maximum of 367 nm for Disponil FES 61 (50 EO units). Further, Zertuche [56] reported that for the mini-EP of St in the presence of divinyl benzene and 1% acrylic acid, using Hitenol BC 10 and 30 as polymerizable nonionic surfactants, *D*_p_ average values of 54.5 and 61.9 nm were recorded, respectively, for latexes with the same polymer content (21.6% *w*/*w*).

#### 3.1.2. Methyl Methacrylate (MMA)

From Appendix A, the global effects of the surfactant type on monomer conversion and *D*_p_ for the EPs of MMA are observed; both responses are significantly affected by the surfactant used to stabilize the emulsions. Monomer conversion with C1EG as stabilizer was 96.14% ± 1.86%, whereas it was 92.11% ± 0.05% for DTAB after 2 h of reaction; however, it is worth mentioning that after only 10 min, the average monomer conversion was already 85.7% and 89.8% for DTAB and C1EG, respectively (see Figure 4). Costa et al. reported a high MMA conversion for both surfactants, [C12mim]Cl (1.7 CMC) and DTAB (1.8 CMC); from their plot (Figure 8 in their paper), monomer conversion was estimated as 97% and 99%, respectively, after 15 min of reaction using an initiator (V-50) concentration of 0.013% and a reaction temperature of 80 °C [28]. The difference in the rate of polymerization between the work of Costa et al. [28] and this study can be mainly attributed to the difference in reaction temperature (10 °C difference); although it is worth mentioning that the results cannot be directly compared because the surfactant concentration, [S], used in this work for MMA/DTAB polymerizations was ca. twice that used in the investigation of Costa et al. [28] (0.048 and 0.023 mol L^−1^, respectively, relative to the aqueous phase). Another substantial difference is that the initiator concentration, [I], used in this contribution was considerably higher than that used in the study by Costa et al. [28] (3.96 × 10^−3^ and 6.07 × 10^−3^ mol L^−1^, respectively).

Regarding the monomer conversion evolution for the EP of MMA with ILs, Costa et al. [28] found similar polymerization behavior when using [C12mim]Cl IL and DTAB at 80 °C; after a short induction period (2–3 min), the polymerization proceeded very fast, reaching ca. 95% monomer conversion in 8–10 min. In the present work, EPs stabilized with the C1EG IL reached ca. 90% monomer conversion in 10 min, as pointed out before, although the reaction conditions were different (lower polymerization temperature and higher [S] and [I]).

With respect to the *D*_p_ results obtained with C1EG and DTAB in this work for the EP of MMA, there were also significant differences between the two surfactants: the average *D*_p_ value was 68.75 ± 5.41 nm for C1EG and 54.27 ± 0.48 nm for DTAB. The explanation for this effect is similar to that offered above for the EP of St.

### 3.2. Molecular Weight

Although not analyzed via ANOVA, it is also evident from Table 1 that the number average molecular weight (*M*_n_) of the polymers obtained with C1EG as the surfactant (102,000–34,000 Da) were lower than those generated with DTAB (233,000–383,000 Da). This can be ascribed to the lower number of particles (*N*_p_) generated with the C1EG surfactant (1.20–1.49 × 10^18^ mL^−1^) as compared to those produced with DTAB (1.65–2.13 × 10^18^ mL^−1^). If it is assumed that the system behaves as a 0–1 system, then upon entry of a second radical into a propagating polymer particle, there is instantaneous termination. Based on this assumption, having a smaller *N*_p_ (C1EG case), each polymer particle will have a higher probability of capturing a second radical, which will shorten the average life of the radicals in the particles, and as a consequence, the molecular weight of the generated polymer will be lower. The lower *N*_p_ in the case of the C1EG surfactant is also consistent with the observed higher *D*_p_.

### 3.3. Contact Angle of PSt Films

The CAs of water in unpurified PSt films cast from C1EG and DTAB stabilized dispersions were 17.3° ± 0.4° and 51.2° ± 0.5°, respectively, whereas for partially purified PSt films, the values were 39.3° ± 1.5° and 73.0° ± 1.7°, correspondingly. All these results are below the values reported by Li et al. of 86.0° ± 2.0° [57] and by Kwok et al. of 88.41° ± 0.28° [58] for pure PSt cast films. The large difference in CA values found here indicates the effect of large quantities of surfactant remaining in the polymer. However, for some applications, it is desirable to have free surfactant, as pointed out before. On the other hand, CAs of water on PSt-cast films were 90.5° ± 1.1° and 90.3° ± 1.2° for the surfactant-free samples derived from C1EG and DTAB stabilized dispersions, respectively. These values are in good agreement with those reported in the literature.

### 3.4. Simulations

POLYRED simulations were run for a few illustrative cases to explore how well the 0–1 model assumption describes the EPs performed with C1EG and DTAB in this investigation. For all the POLYRED simulations, the required kinetic and physico–chemical parameters were taken from the literature; a list of the main parameters used and their corresponding sources are given in Appendix A. Figure 5 shows the evolution of the calculated value of the average number of radicals per particle $\bar{n}$ with monomer conversion for four of the performed EPs: experiments 15, 28, 30, and 32 (hereafter referred to as E15, E28, E30, E32, etc.). In all cases, $\bar{n}$ is lower than 0.5 across most of the monomer conversion range; at very high monomer conversions (>95%), the gel effect might take place and decrease the rate of termination. This further supports the assumption made in the previous paragraph that instantaneous termination might occur upon the entry of a second radical into a propagating polymer throughout the entire polymerization. In the range 0 < $\bar{n}$ < 0.5, bimolecular termination is not rate-determining [59], and the value of $\bar{n}$ is mainly governed by the interplay between the entry/exit of radicals into/from polymer particles. Radical exit is dominant as $\bar{n}$ approaches 0, whereas radical entry dominates when $\bar{n}$ ~ 0.5.

### 3.5. Factorial Analysis and Discussion of Main Effects

Due to the multifactorial nature of the experiments, they can be analyzed within different categories. Originally, the experimental design was meant to be a full factorial design, but this could not be achieved due to experimental limitations; however, some fractions of the full design were isolated (four fractions shown with different shadows/colors in Table 1), and pure (main) effects could still be estimated at fixed levels of other variables. Within these groups, a high level of surfactant (C1EG or DTAB) or of initiator (V-50) are indicated with a (+) sign to the right of the numerical value, and a low level with a (−) sign. Within the analyzed groups, three levels were used for surfactant, with the highest level indicated with a (++) sign. Although the numerical values grouped in a given level may not be identical due to the final experimental variations, they are grouped together if their values did not differ more than 5%. This is the case, for example, for the two low (−) values of initiator V-50 for E25 and E28, which were grouped in the same level although their specific values were 3.12 and 3.16 × 10^−4^ mol, respectively (a difference of ~1.3%).

### 3.6. Surfactant and Initiator Concentration Effects on EPs of St with C1EG and DTAB

The statistical assessment of the overall effect of surfactant and initiator concentration on St conversion and *D*_p_ is shown in Appendix A for C1EG, which includes the respective set of experiments (E15, E21, and E34). Similarly, results when DTAB was used as surfactant are shown in Appendix A for the corresponding set of experiments (E19, E25, and E28).

#### 3.6.1. Surfactant Effects

C1EG: According to the ANOVA in Appendix A, the increase from low to high levels of surfactant (7.7 to >10 × CMC) leads to a substantial increase in monomer conversion, whereas the effect on D_p_ is not significant, probably due to the confounding effect of initiator concentration. However, a closer look at the block of experiments (E15, E21, and E34) and their replicates (upper part of Table 1) allows one to isolate and analyze some nearly pure effects: E15 and E34 show the effects of surfactant concentration (high for E15 and low for E34) at a low level of initiator concentration. As expected, E15 shows higher *N*_p_, higher monomer conversion, and lower *D*_p_ than E34, all of which are consistent with the standard theory of EP. In fact, taking Equation (3), it is possible to estimate β by assuming that the initiator concentration is the same. Equation (3) is simply derived from the Smith–Ewart expression for *N*_p_ in EP, in which *α* = 0.4 and *β* = 0.6 [50].
(3)$$\frac{N_{p1}}{N_{p2}}={\left(\frac{{\left[I\right]}_{1}}{{\left[I\right]}_{2}}\right)}^{\alpha }{\left(\frac{{\left[S\right]}_{1}}{{\left[S\right]}_{2}}\right)}^{\beta }$$

In our investigation, an experimental value of 0.54 was obtained for β, which is close to the theoretical one of 0.6 given by the Smith-Ewart theory. To study this effect further, reactions E15 and E34 (high and low levels of surfactant, respectively) were simulated with POLYRED, and the obtained results for monomer conversion (simulated and experimental) are compared in Figure 6. The higher monomer conversion attained for increased surfactant concentration is expected for any EP system according to the traditional Smith-Ewart theory [50], which was confirmed by the POLYRED simulations.

The pair of experiments E21 and E34 should show the effects of initiator concentration (high for E21 and low for E34) at a low (−) level of surfactant. The effect on monomer conversion is opposite to what is expected (<3% higher for E34), but this difference is not statistically significant according to the ANOVA in Appendix A. On the other hand, *N*_p_ shows the expected trend (24% higher for E21). Usually, a higher *N*_p_ leads to higher monomer conversion if the rest of the factors are equal. This apparent inconsistency is ascribed to a more pronounced gel effect in E34 corresponding to a higher polymer molecular weight, noticing that monomer conversion for this reaction, as compared to E21, becomes higher only during interval III of the EP.

DTAB: In the case of DTAB (experiments E19, E25, and E28), the ANOVA in Appendix A shows that the highest surfactant concentration (3.5 × CMC) leads to an improvement in monomer conversion compared with the other two lower concentrations (2 and 3 × CMC), although this difference turns out to be not statistically significant at a *p*-level of 0.05, apparently due to the large variability in monomer conversion of E25 (see Appendix A). For a more detailed analysis, comparisons between the pairs of experiments E19/E25 and E19/E28 were made (in both cases E19 had higher levels of surfactant and initiator) and are consistent with the higher monomer conversion exhibited by E19 as compared to the other two. An unexpected outcome is the lower *N*_p_ exhibited by E19 compared to E25, but this observation is discussed below with the aid of simulations. Note that *N*_p_ is a parameter calculated based on monomer conversion and the average particle diameter, and that the monomer conversion of E25 shows rather large variability, which introduces some additional uncertainty to these particular results. Regarding experiments E25 and E28, the former shows similar monomer conversion and higher *N*_p_ than the latter, which can be attributed to the higher [S] used in E25.

Further analysis exhibits good agreement between POLYRED simulations and experiments, as shown in Appendix A for reactions E19 and E28 for high and low levels of surfactant DTAB, respectively. Although the effects on *D*_p_ are not easy to explain due to the combined effects of surfactant and initiator concentrations, POLYRED simulations are in good agreement with the experimental results, at least in qualitative terms at moderate monomer conversions, and quantitatively at high monomer conversions. 

Reactions E19 and E25 were also simulated with POLYRED and compared to experimental results for a better understanding of the combined effect of initiator and surfactant concentrations. The results displayed in Appendix A for both experiments and simulations show that monomer conversion is consistently higher for E19 than for E25. Although *D*_p_ values are experimentally and theoretically very similar for both reactions, with up to ~90% monomer conversion, since E19 reaches higher monomer conversion than E25, its final particle size is also higher, which results in lower *N*_p_. Note, however, that if comparison of the *N*_p_ values is made at similar monomer conversions (~90%), the estimation of *N*_p_ for E19 is higher than that of E25 since the *D*_p_ values are about the same in both cases. This could be explained by the occurrence of some coagulation in the last stages of E19 (Appendix A). The results for *D*_p_ show that there are no significant differences attributed to initiator concentration for any of the two surfactants, which suggests that initiator effects may be confounded with surfactant concentration effects.

In general, when the effect of a change in a response variable is expected by theory, but the difference observed in the response is not statistically significant, there are several factors or a combination of factors that can explain the apparent contradiction: (i) the *p*-value used in statistical analysis might be too strict; (ii) the selected magnitude of change on the (independent) variable is not large enough; or (iii) the experimental error is too large. However, the lack of statistical significance does not mean that the effect is not present; it simply means that it could not be evidenced by the utilized experimental conditions. Therefore, a combination of models (simulation and theory) and statistical tests complement each other in the analysis.

#### 3.6.2. Initiator Effects

Appendix A shows the effect of initiator on St conversion in the presence of C1EG, indicating no significant difference between V-50 concentrations of 3.68 and 5.53 × 10^−4^ mol for a constant C1EG concentration (7.7 × CMC). There are, though, significant differences in monomer conversion when C1EG is 10.9 × CMC; even though in this case a relatively low level of initiator was used, monomer conversion was significantly higher compared to the other two experiments. This suggests that under the investigated conditions, the surfactant concentration effect overrules the initiator concentration effect. The corresponding reactions, E15 and E21, were simulated with POLYRED to further study the combined effect of initiator and surfactant concentrations, and the results (Figure 7) show that the surfactant effect dominates the initiator effect; at higher surfactant concentration, monomer conversion is higher, and *D*_p_ is smaller at the same monomer conversion. The final *D*_p_ value of E15 is higher because it corresponds to higher monomer conversion than that in E21 (96.9 vs. 85.9%, respectively), which allows particles to grow larger.

In the case of DTAB as surfactant (Appendix A), monomer conversion is significantly different for the relatively high concentration of initiator (3.93 × 10^−4^ mol) as compared to the experiments at the lower level of initiator (3.14 × 10^−4^ mol), although there is some confounding (in the statistical sense) with the surfactant concentration effect since the experiment with high initiator level was also run with the highest surfactant concentration (3.5 × CMC) as compared to the other two (2 and 3 × CMC). In any case, these results are also in agreement with classical EP theory [50,51].

### 3.7. Surfactant and Initiator Concentration Effects on MMA EPs with C1EG

The overall effects of surfactant and initiator concentrations on MMA conversion and *D*_p_ are exhibited in Appendix A for C1EG, including the block of experiments (E27, E30, and E36). When DTAB was used as stabilizer, only one experiment was performed (E32).

#### Surfactant Effects

When a high level of surfactant (10.1 × CMC) was used in the case of C1EG (Appendix A), as expected, monomer conversion was higher than that obtained for a lower surfactant concentration (7.7 × CMC). In particular, experiments E27 and E36 reflect the pure effect of surfactant concentration (high for E27 and low for E36, with both at a low level of initiator concentration) on monomer conversion and *N*_p_. As expected, both responses show higher values for E27 than for E36, although the differences are not statistically significant at the *p*-level used in the corresponding ANOVA (Appendix A). This pair of experiments is analyzed further with the aid of POLYRED simulations (Figure 8), which show good agreement between the model and the experimental results for monomer conversion and *D*_p_, confirming the expected effect of surfactant on monomer conversion. Regarding *D*_p_, the effect observed in this group of reactions seems to be weakly correlated with surfactant concentration and, rather, linked to initiator effects.

On the other hand, comparison of experiments E30 and E36 allows one to assess the effect of initiator concentration (high for E30; low for E36). No significant initiator concentration effects are found in Appendix A (C1EG) for monomer conversion. However, for *D*_p_, a lower level of initiator (3.68–3.80 × 10^−4^ mol) leads to a significantly higher average particle size compared to that corresponding to a higher initiator concentration. The smaller *D*_p_ of the latter is consistent with a larger *N*_p_ (see Table 1), which is expected for a higher value of [I] and consistent with classical Smith–Ewart theory. This is confirmed by simulations of reactions E30 and E36, as shown in Figure 9 with model predictions that describe well the experimental data.

The previous effects can be better understood by considering the expression for the rate of polymerization in an EP [51]:(4)$$R_{p}=\frac{k_{p}\;N_{p}\bar{n}{\left[M\right]}_{p}}{N_{A}}$$
where $R_{p}$ is the rate of polymerization (mol L^−1^ s^−1^), $k_{p}$ is the propagation rate coefficient (L mol^−1^ s^−1^), $N_{p}$ is the number of particles per liter of water (L^−1^), $\bar{n}$ is the average number of radicals per particle, ${\left[M\right]}_{p}$ is the monomer concentration in the particles (mol L^−1^), and $N_{A}$ is Avogadro’s number. From Equation (4), an increase in $N_{p}$, which itself is proportional to fractional powers of [I] and [S], leads to an increase in $R_{p}$ and therefore to higher monomer conversion in equal reaction times, assuming the rest of the variables remain constant.

Finally, there are no experiments to compare with E32, in which DTAB was used as a surfactant to polymerize MMA.

### 3.8. Some Considerations on the Potential Application of ILs as Surfactants

ILs are considered “green” because their vapor pressure is negligible; however, their toxicology data have been investigated in a very limited way [60]. Although ILs practically do not evaporate, and therefore do not cause air pollution, this does not mean that they will not harm the environment if they enter aquatic systems. Most ILs are water-soluble and can enter aquatic environments through spills or by accident. It is known that many commonly used ILs (e.g., 1-*N*-butyl-3-methylimidazolium hexafluorophosphate, [bmim] [PF_6_] and 1-butyl-3-methylimidazolium tetrafluoroborate, [bmim] [BF_4_]) decompose in the presence of water, forming hydrofluoric and phosphoric acids as a result [61,62]. Toxicity and ecotoxicity information, which describes the metabolism and degradability of ILs, has been reviewed by Egorova et al. [60]; this contribution also summarizes many applications of ILs, such as in chemical and catalytic reactions, biotechnology, electrochemistry, and many others. On the other hand, it is conceivable that in cases where a dry polymer is the target of a process, after the polymer has been recovered from EP via coagulation and separation, water and IL can be separated by flash distillation given the almost null vapor pressure of the latter. This would allow water and/or IL to be reused in the same process (and in a circular approach) after readjusting the corresponding emulsion formulation. Another advantage of using ILs as surfactants is that, depending on their chemical structure and film-formation conditions, they offer the possibility of manipulating the hydrophiliciy/hydrophobicity of films derived from the corresponding latexes, as demonstrated by Rozik et al. [38] and by the data on contact angles reported in this work. Furthermore, as demonstrated before, the use of ILs as surfactants/stabilizers in heterogeneous polymerizations enables the possibility to modulate the size of generated polymer particles from the nano- to the milli-meter range [26]. Further research with other ILs and monomers is certainly needed to detect additional advantages.

## 4. Conclusions

EPs of St and MMA were successfully performed and stabilized with C1EG at 70 °C using V-50 as the free-radical initiator. Monomer conversion reached after two-hour reactions was relatively low (around 70%) when the surfactant C1EG concentration was 2–3-fold its CMC, and therefore the concentration was increased up to 7.7–10.9 × CMC, values at which monomer conversion was above 90% after two hours of reaction, comparable to that obtained with the conventional surfactant DTAB at concentrations of 2–3-fold its CMC. 

When compared with the work of Costa et al. using [C12mim]Cl IL for MMA EP, even though the reaction conditions used in that work and this study are not directly comparable, they are qualitatively similar, confirming the viability of using either of these ILs as surfactants in EPs.

In general terms and confirmed by the experimental trends and POLYRED simulations, the EP stabilized with IL C1EG seems to follow the classical Smith–Ewart theory, at least in qualitative terms. With the exception of a few experiments that exhibit some significant deviations from POLYRED simulations, the model implemented in this software correlates well with the obtained experimental results.

## Figures and Tables

**Figure 1 polymers-14-03475-f001:**
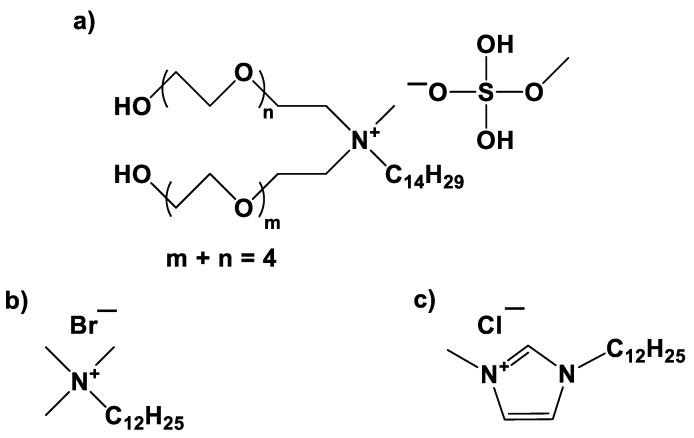
Molecular structures of (**a**) IL IOLILYTE C1EG, (**b**) DTAB, and (**c**) IL [C12mim]Cl.

**Figure 2 polymers-14-03475-f002:**
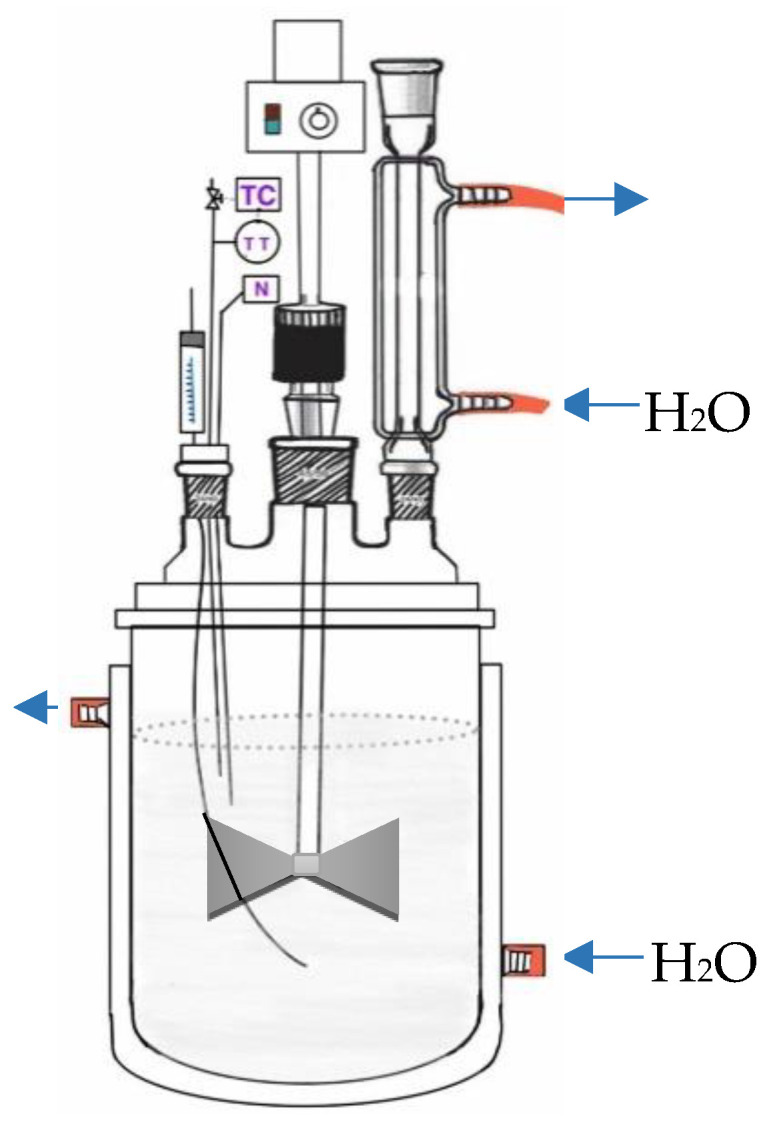
Schematic representation of the EP reactor.

**Figure 3 polymers-14-03475-f003:**
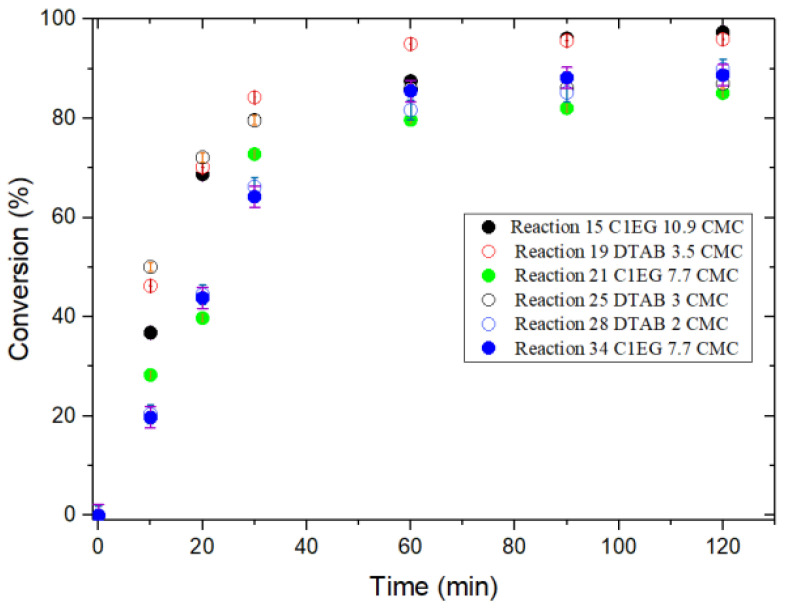
Evolution of monomer conversion for EP of St with IL C1EG and DTAB as emulsion stabilizers at 70 °C.

**Figure 4 polymers-14-03475-f004:**
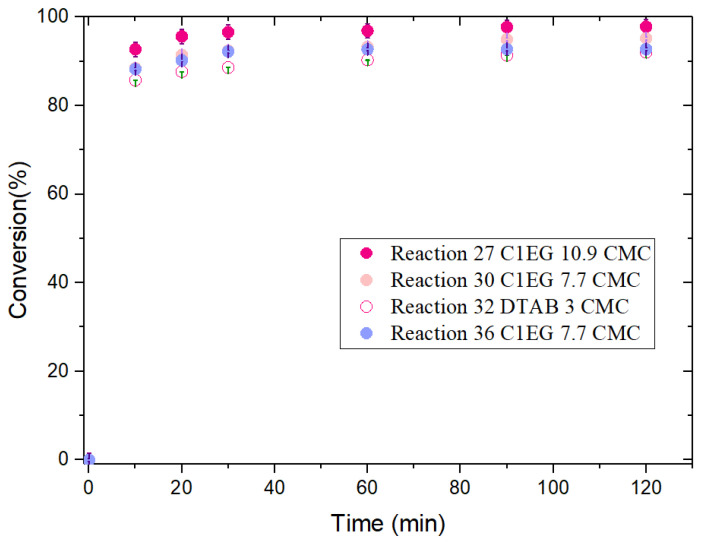
Evolution of monomer conversion for the EP of MMA with IL C1EG and DTAB as emulsion stabilizers at 70 °C.

**Figure 5 polymers-14-03475-f005:**
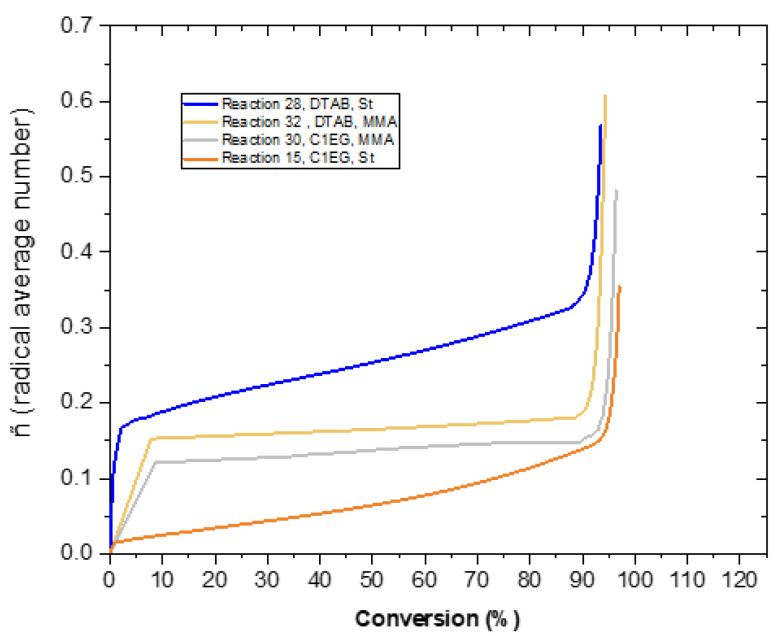
Evolution of the average number of radicals in polymer particles ($\bar{n}$) with monomer conversion calculated by simulation (POLYRED) for four selected EPs: E15, E28, E30, and E32. The experimental formulation used in the EPs is summarized in Table 1, whereas the parameters used in the simulations are displayed in Appendix A.

**Figure 6 polymers-14-03475-f006:**
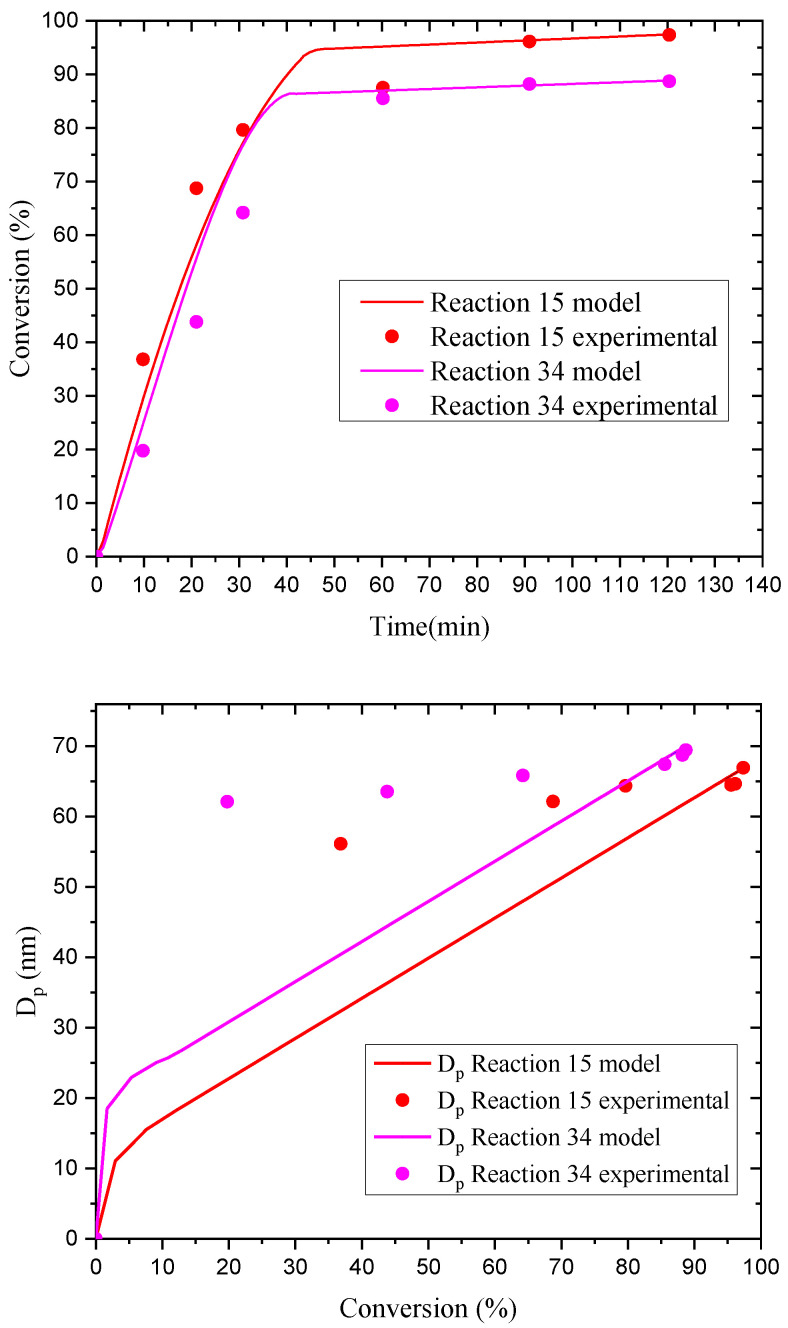
Comparison of simulation (POLYRED) and experimental data for monomer conversion vs. reaction time (**top**) and *D*_p_ vs. monomer conversion (**bottom**) for experiments E15 and E34.

**Figure 7 polymers-14-03475-f007:**
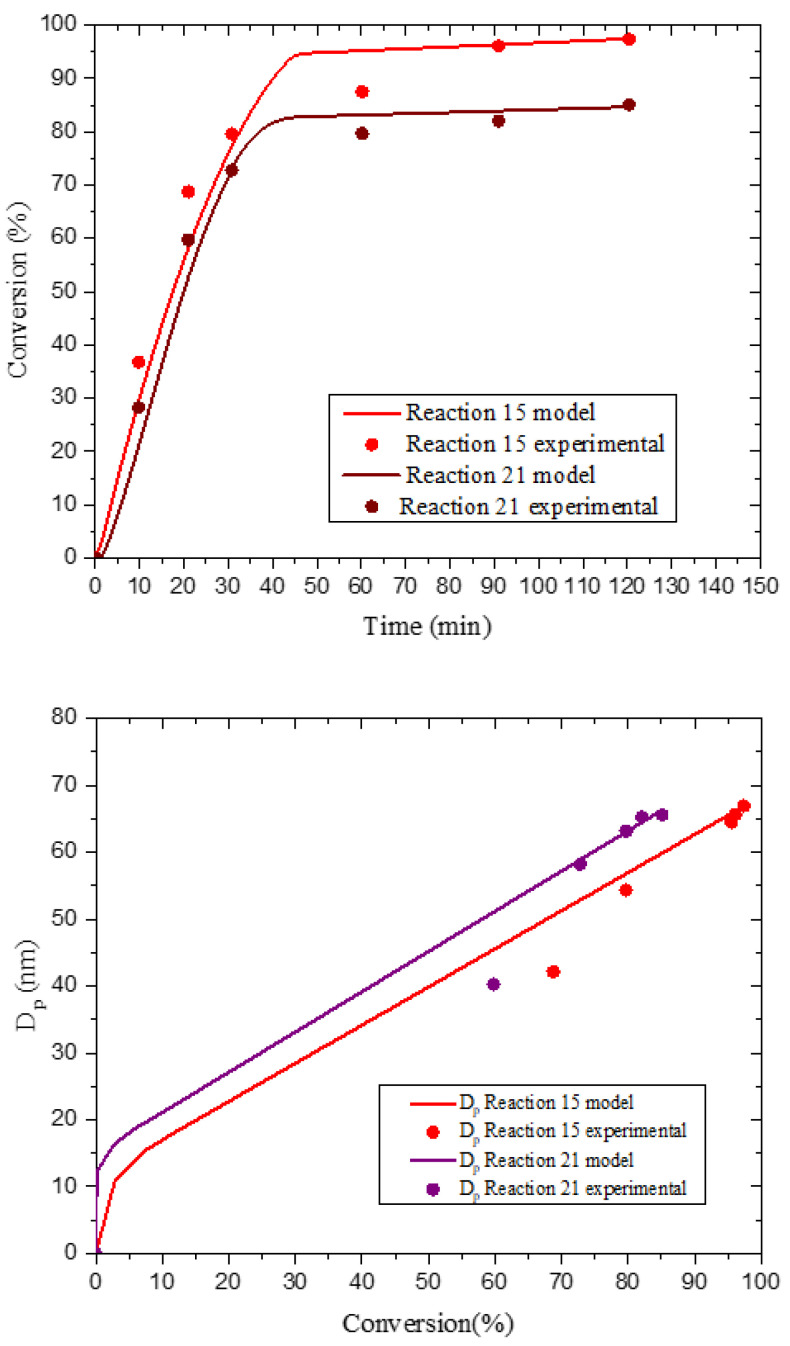
Comparison of simulation (POLYRED) and experimental data for monomer conversion vs. reaction time (**top**) and *D*_p_ vs. monomer conversion (**bottom**) for experiments E15 and E21.

**Figure 8 polymers-14-03475-f008:**
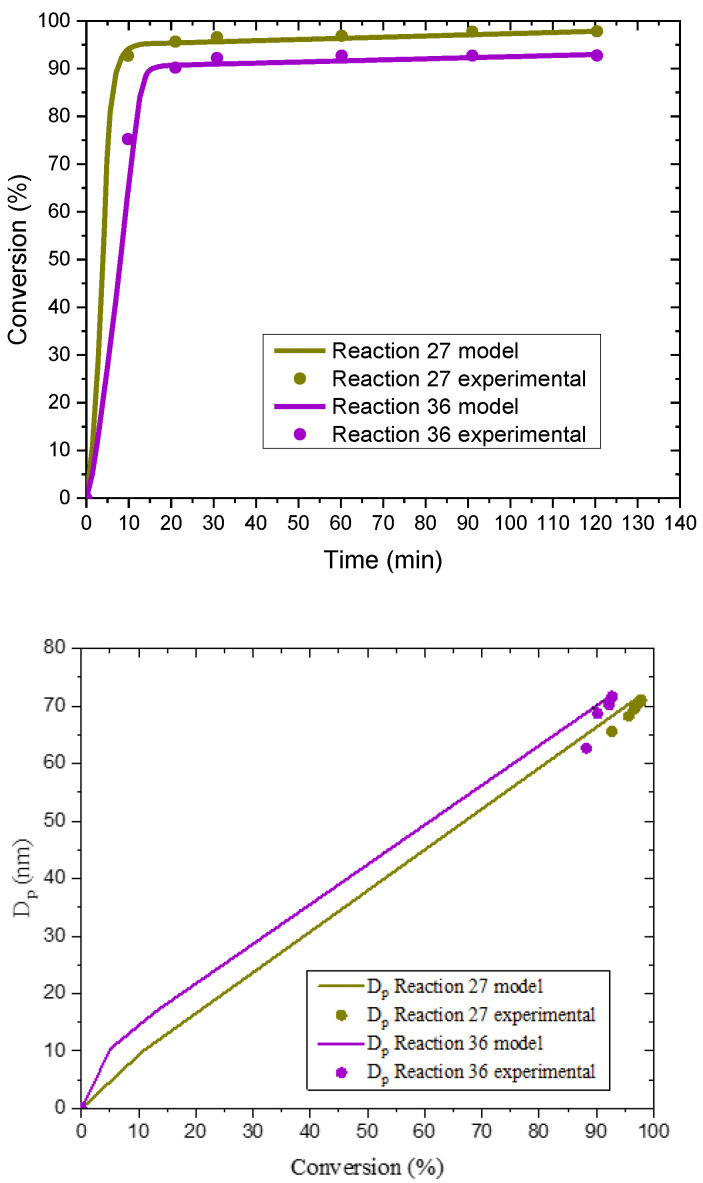
Comparison of simulation (POLYRED) and experimental data for monomer conversion vs. reaction time (**top**) and *D*_p_ vs. monomer conversion (**bottom**) for experiments E27 and E36.

**Figure 9 polymers-14-03475-f009:**
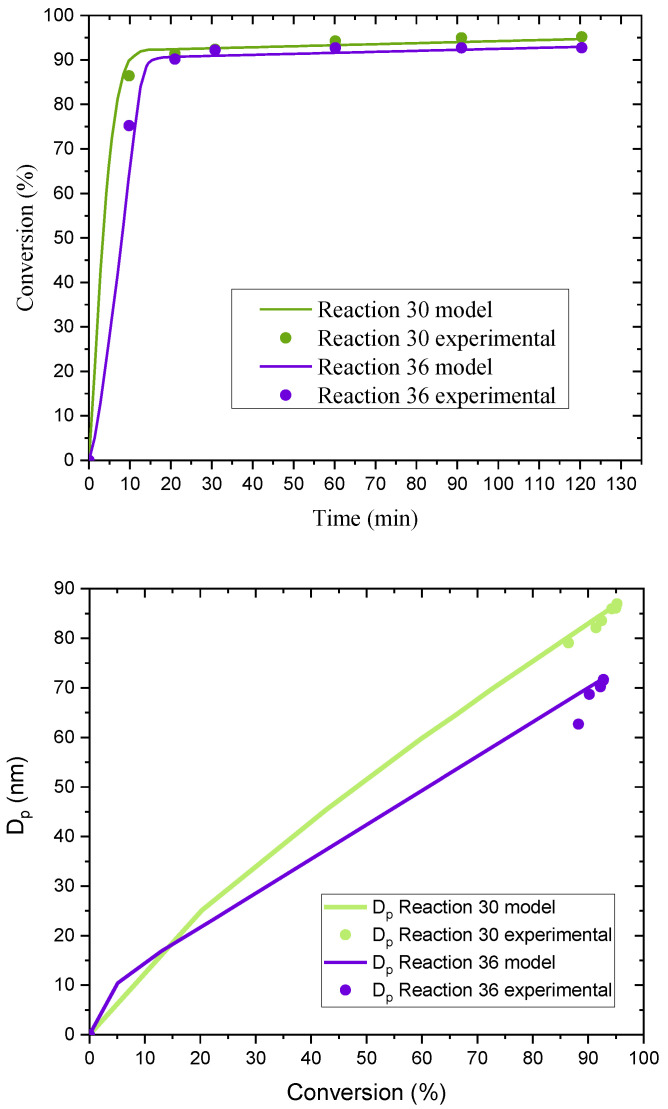
Comparison of simulation (POLYRED) and experimental data for monomer conversion vs. reaction time (**top**) and *D*_p_ vs. monomer conversion (**bottom**) for experiments E30 and E36.

**Table 1 polymers-14-03475-t001:** Experimental design and main responses of the EPs carried out at 70 °C. All quantities in the recipe are in moles (columns 2–7) except for water. S = surfactant; I = initiator. The groups of reactions (in different colors) correspond to fractions of a factorial experimental design for a fixed monomer and type of surfactant.

Reaction	MMA × 10^1^	St × 10 ^1^	S = C1EG × 10^3^	S = DTAB × 10^3^	I = V50 × 10^4^	Water [g]	*D*_p_ [nm]	Conversion [%]	*M*_n_ × 10^−3^ [Da]	Ð	*N*_p_ × 10^−18^ [mL^−1^]
15	0	1.87	2.32 ^a1^ (+)	0	3.85 (−)	78.8	66.21 ± 1.1	96.9 ± 0.6	124	6.5	1.44 ± 0.03
21	0	1.94	1.69 ^b1^(−)	0	5.53 (+)	79	64.26 ± 1.2	85.9 ± 1.8	102	5.8	1.49 ± 0.07
34	0	1.92	1.66 ^b1^(−)	0	3.68 (−)	78.8	69.6 ± 0.2	88.2 ± 0.7	234	3.8	1.20 ± 0.03
19	0	1.88	0	4.47 ^b^ (++)	3.93 (+)	78.8	59.2 ± 0.5	96.4 ± 0.7	233	4.6	2.13 ± 0.003
25	0	1.68	0	3.78 ^a^ (+)	3.12 (−)	78.8	53.7 ± 0.9	89.4 ± 3.5	338	4.6	2.42 ± 0.16
28	0	1.6	0	2.51 ^c^ (−)	3.16 (−)	78.8	61.2 ± 0.1	90.2 ± 0.4	383	4.0	1.65 ± 0.19
27	1.94	0	2.15 ^c1^ (+)	0	3.8 (−)	78.7	71.5 ± 0.9	97.6 ± 0.3	383	4.0	1.32 ± 0.06
30	1.91	0	1.66 ^b1^ (−)	0	5.57 (+)	79.2	60.90 ± 0.4	95.2 ± 0.01	336	3.8	2.08 ± 0.06
36	1.96	0	1.66 ^b1^(−)	0	3.68 (−)	79.2	72.5 ± 0.3	94.9 ± 3.0	300	4.7	1.26 ± 0.05
32	1.59	0	0	3.79 ^a^	3.13	78.7	54.3 ± 0.5	92.1 ± 0.1	533	3.7	2.44 ± 0.13

Abbreviations: *D*_p_ = particle diameter; *M*_n_ = number average molecular weight; Đ = dispersity of the molecular weight distribution; *N*_p_ = number of particles per mL of water. ^a^ CMC × 3, ^b^ CMC × 3.5, ^c^ CMC × 2, ^a1^ CMC × 10.9, ^b1^ CMC × 7.7, ^c1^ CMC × 10.1.

## Data Availability

Not applicable.

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
