# Peer review of "Emulsion Polymerization Using an Amphiphilic Oligoether Ionic Liquid as a Surfactant"

_polymers, 2022, doi:10.3390/polym14173475_

Round 1

Reviewer 1 Report

In their manuscript, Jimenez-Victoria present a very thorough study on the emulsion polymerization of styrene and MMA with a conventional surfactant and an ionic liquid. 

Polymerization kinetics are mathematically modelled, giving a coherent picture. 

Indeed there are only few points to be considered before publication:

- the prospect of a comparison to the results with C12mimCl is somehow misleading - the reader may expect experiments and not just literature values. Pleas clarify

- Table 1: please check the powers of Mn and Np. Negations does not seem appropriate. In addition the subscript says "...particles per L...", the unit in the table reads mL-1. Chose one

- the method for evaluating the conversion is missing in the experimental part

- please add one convincing sentence about the use of IL - one of the advantages, the virtual absence of vapor pressure, seems obsolete when using in aqueous solution. 

To summarize, there are only minor issues to be addressed before the manuscript can be accepted

Reviewer 2 Report

This work has a potential and my comments are as follows:

- First of all, the Abstract should contain answers to the following question: why is this investigation important?

The Introduction should make a compelling case for why the emulsion polymerization is beneficial with a clear statement of its novelty and originality by providing relevant information and providing answers in connection with the ionic liquid.

- What are the special cases of your study?

- Article needs proofreading to eliminate minor typos.

- Punctuation is missing after some equations.

- For enhancing the introduction section with the new publications, old references may be replaced with new ones such as:

Dynamism of a hybrid Casson nanofluid with laser radiation and chemical reaction through sinusoidal channels

Computational Framework of Magnetized MgO–Ni/Water-Based Stagnation Nanoflow Past an Elastic Stretching Surface: Application in Solar Energy Coatings

Reviewer 3 Report

This is a well-written manuscript and could be published based on the importance of ionic liquid for various areas. I only have a minor suggestion, that, what is the advantage of ionic liquid surfact in emulsion Polymerization. It seems there is not difference from ionic liquid surfactant and other conventional surfactants. Please highlight the importance of ionic liquid surfactant in polymerization.
